# HIV testing and seroprevalence among couples of people diagnosed with HIV in China: A meta-analysis

Ci Zhang[1,2], Han-Zhu Qian[1,3], Xi Chen[4], Scottie Bussell[5], Yan Shen[1,2], Honghong Wang[1,2]*, Xianhong Li[1,2]*

1 Xiangya School of Nursing, Central South University, Changsha, Hunan Province, China, 2 Xiangya Center for Evidence-Based Nursing Practice & Healthcare Innovation (A JBI Affiliated Group), Changsha, Hunan Province, China, 3 School of Public Health, Yale University, New Haven, Connecticut, United States of America, 4 Hunan Provincial Central for Disease Control and Prevention, Changsha, Hunan Province, China, 5 Department of Health and Human Services, Parker Indian Hospital, Parker, Arizona, United States of America

* xianhong_li228@hotmail.com (XL); honghong_wang@hotmail.com (HW)

**Data Availability Statement:** All relevant data are within the manuscript and its Supporting information files.

## Abstract

### Background

Partner notification and testing could expand HIV testing and link infections to care. We performed a meta-analysis on HIV testing rate and prevalence among couples of people diagnosed with HIV in China.

### Methods

Six electronic databases (PubMed, Cochrane Library, Embase, Web of Science, the China National Knowledge Internet, and WanFang) and abstracts of five HIV/sexually transmitted infections conferences were searched up to February 1, 2020. Meta-analysis was conducted using a random-effects model to assess HIV testing rate and prevalence among couples of Chinese people diagnosed with HIV.

### Results

Of 3,657 records retrieved, 42 studies were identified. Among them, three studies were conducted among pregnant women and 10 among men who have sex with men. The pooled uptake rate of couples HIV testing among Chinese people diagnosed with HIV was 65% (95% confidence interval, 57% -73%; 23 studies). The pooled HIV prevalence among couples who had an HIV test was 28% [24%-32%] (38 studies). Subgroup analyses showed that the pooled couples HIV testing uptake rates among pregnant women and men who have sex with men were 76% [66%-86%] (3 studies) and 49% [30%-68%] (8 studies), and the pooled HIV prevalence in two populations was 53% [27%-78%] (3 studies) and 14% [10%-17%] (10 studies), respectively.

**Funding:** This study was funded by Central South University Innovation-driven project (XL; 2018CX036; http://www.csu.edu.cn/) and National Natural Science Foundation of China (XL; 72074226; https://isisn.nsfc.gov.cn/egrantweb/). The funders had no role in study design, data collection and analysis, decision to publish, or preparation of the manuscript.

**Competing interests:** The authors have declared that no competing interests exist.

## Conclusions

Nearly two-thirds of couples of people diagnosed with HIV have had an HIV test, of whom 28% were positive. Couples of MSM with a positive HIV diagnosis had a lower testing rate, which indicates more effective strategies need to be carried out to improve couples HIV testing among Chinese MSM.

## Introduction

Historically, intravenous drug use played a major role in HIV transmission in China; however, currently, sexual intercourse is the main mode of HIV transmission [1, 2]. Only 68% of people living with HIV (PLWH) are aware of their positive status [3], which is well below from the 90% awareness target by 2020 as set by the Joint United Nations Programme on HIV and AIDS [4]. This suggests that there is still a significant gap of HIV testing in China.

Couples HIV Testing (CHT) is an approach to have couples tested for HIV and promote HIV testing [5]. This strategy can increase their knowledge of their serostatus and encourage disclosure among people who are in an ongoing sexual relationship [6, 7]. Studies show that CHT is a feasible strategy that can expand HIV testing and further, prevent HIV transmission by increasing condom use among discordant couples [8, 9]. In addition, encouraging CHT among people who have been diagnosed with HIV (PDWH) can identify additional HIV-infected individuals and direct them to early antiretroviral therapy (ART) [10].

The World Health Organization (WHO) released guidelines for the testing and counseling of couples in 2012 and strongly recommended CHT as an essential strategy to promote HIV testing and reach more PLWH [7]. In the 13th Five-Year Plan, which mapped out the tasks to build a healthy China, the Chinese government encouraged CHT among PDWH [11]. We conducted a meta-analysis examining CHT uptake rate and HIV prevalence among Chinese PDWH in order to provide a useful summary of evidence on CHT practice and outcomes.

## Materials and methods

This meta-analysis was reported according to the PRISMA guidelines (S1 Table) [12].

### Inclusion criteria

We defined CHT as 1) HIV testing of sexual partners (whether married or not) reported by participants; and 2) testing recorded by the Chinese Centers for Disease Control (CDC) staff. The target population was defined as Chinese PDWH who had documented records of CHT.

Studies were eligible if they reported data on at least one of the following outcomes: the proportion of CHT among Chinese PDWH, and the HIV prevalence among CHT for Chinese PDWH. CHT uptake rate was calculated using the following formula: (number of PDWH couples who had HIV testing) / (number of PDWH). HIV prevalence among CHT was calculated using the following formula: (number of infected couples)/(number of couples who had HIV testing). Randomized controlled trials (RCTs), quasi-experimental studies, and observational (cross-sectional, cohort, and case-control) studies were eligible for inclusion. For experimental trials and cohort studies, baseline data were used for the meta-analysis. Studies were excluded if they were qualitative, a review, or a duplicate report.

## Search strategy

We searched six electronic databases (PubMed, Embase, Web of Science, Cochrane Library, the China National Knowledge Internet, and WanFang) and abstracts from the International AIDS Society (IAS), HIV Diagnostics Conference (HDC), Canadian Association of HIV Research (CAHR), Infectious Diseases Society of America (IDSA), and the International Congress of Behavioral Medicine (ICBM) for publications up to February 1, 2020. Our search terms included (China OR Chinese) AND (("couple HIV testing" OR "couples HIV testing" OR "partner HIV testing" OR "partner testing" OR "couple testing" OR "couples testing") OR ((test OR testing) AND ("couple" OR "couples" OR "partner" OR "partners"))) AND "HIV Infections"[MeSH] OR "HIV infections" OR "HIV infection" OR "Acquired Immunodeficiency Syndrome"[MeSH] OR "Acquired Immunodeficiency Syndrome" OR "Acquired Immunodeficiency Syndromes" OR AIDS OR HIV). The search was limited to human studies, and English and Chinese language publications. We included abstracts if full texts could not be accessed, and we contacted the authors for original data if needed. Gray literature was screened using Google Scholar. In addition, the reference lists of included studies and previously published reviews were searched for additional potentially eligible studies. The literature search and study selection procedures are described in Fig 1.

## Data screening and extraction

Two reviewers (C.Z. and Y.S.) independently screened the titles and abstracts of the articles referring to Chinese PDWH and CHT, and then screened full texts for eligibility. Discrepancies (about 5%) were resolved by discussions with a third reviewer (X.L.). A standard data extraction form was used to extract the variables (including author, year, province, study design, study period, sample size, population, type of union, outcome measurement, CHT uptake, and HIV prevalence among CHT) from the identified studies.

## Quality assessment

We performed a quality assessment of the included studies using the Joanna Briggs Institute Critical Appraisal Checklist for cross-sectional studies [13]. RCTs, quasi-experimental studies, cohort studies, and case-control studies were also evaluated using the cross-sectional study checklist, as data was only extracted from the baseline phase. The checklist has eight items, so the total score for each study ranged from 0 to 8, and it was categorized as low quality $\leq 3$, moderate = 4–6, and high quality $\geq 7$. The quality assessment was conducted by two independent reviewers (C.Z. and Y.S.), and the disagreements were resolved by discussion with a third reviewer (X.L.).

## Statistical analysis

STATA 12.0 was used to summarize the results. A meta-analysis was conducted using the DerSimonian-Laird random-effect model to produce pooled proportions and 95% confidence intervals (95% CI) [14, 15]. Heterogeneity was assessed using $I^2$ statistics. In addition, subgroup analyses were performed to explore the source of heterogeneity by the study population and the type of couples. In this study, we classified couples into two types: couples that were defined as people in an ongoing sexual relationship and spouses that were defined as people who have a legal marital relationship. Egger's tests and funnel plot visual inspection were performed to detect publication bias [16]. A sensitivity analysis was performed to detect the impact of each study on the pooled estimate using the leave-one-out approach, which is a repeating procedure of removing one study from the analysis each time. There are two ways to

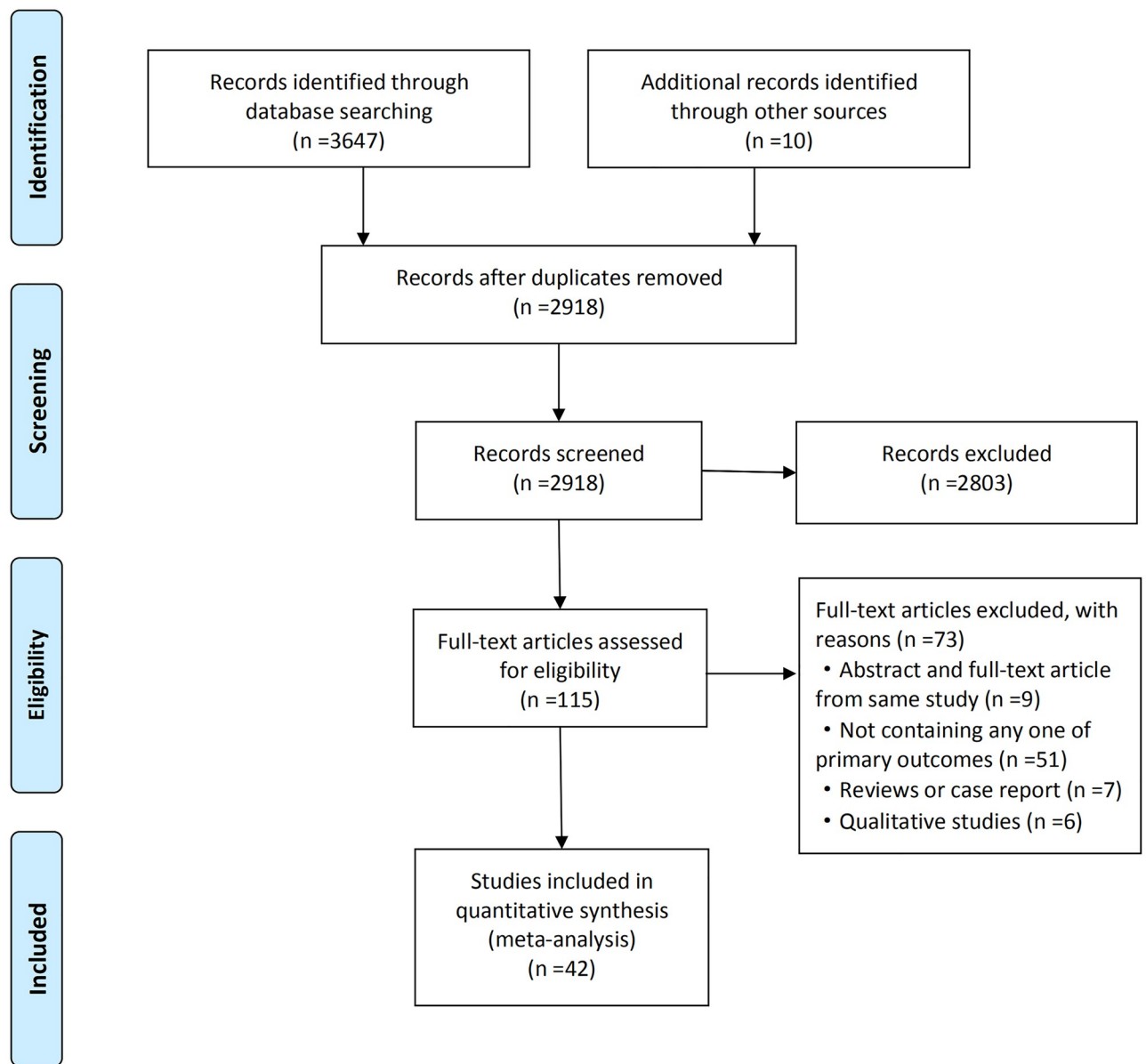

**Fig 1. Flow chart of literature search and selection procedures and outcomes.**

determine that one study impacts the pooled estimate significantly: the estimate after removing this study is across the upper or lower CI limit lines of the pooled estimate; the upper or lower CI limit lines after removing this study are across the pooled estimate line.

## Results

### Description of included studies

A total of 3,657 records were retrieved, of which 3,647 were from database searching and 10 from other sources. After removing duplicates, 2,918 records were identified and 2,803 records were excluded by reviewing the title or abstract. The eligibility of the remaining 115 records

was assessed by reviewing the full-text articles, and 73 studies were excluded. Finally, 42 studies were included in the meta-analysis (Fig 1).

These included studies were conducted between 1989 and 2019 and published during 2009–2019, which are presented in Table 1 [17–58]. Thirty-eight studies were published in Chinese [17–51, 54, 55, 58] and four in English [52, 53, 56, 57]. Geographically, the 42 included studies covered 20 provinces (58.8%) of all 34 provinces in China, with 13 studies conducted in Yunnan Province [17, 18, 21–23, 27, 31, 35, 38, 39, 51, 56, 58] and 7 studies in Guangxi Province [19, 34, 36, 41, 42, 44, 51]. The sample size varied from 28 to 48,931. Most studies were cross-sectional (83.3%, 35/42) [17, 19–22, 25–30, 32–45, 47–55, 58]. The majority of studies were conducted among general PDWH (27/42, 64.3%) [18, 20, 22, 26, 27, 29, 32–52, 54], 11 (23.8%) among men who have sex with men (MSM) [19, 23–25, 28, 30, 31, 52, 55–57], 3 (7.1%) among pregnant women [17, 21, 58], and one (2.4%) among blood transfusion recipients [53]. In most studies (85.7%) [19, 20, 22–33, 35–47, 49–56, 58], the outcomes of CHT uptake and HIV infection were recorded by a CDC staff member when PDWH couples underwent HIV testing in a CDC. Reported outcome variables included uptake of CHT (23 studies) [17–21, 24, 25, 27–31, 33, 34, 39, 40, 44, 46–48, 55, 57, 58] and HIV prevalence among CHT (38 studies) [17, 20–43, 45, 47–58]. The quality of each study was evaluated in detail (S2 Table). Most studies were assessed as high quality (38/42, 90.5%) [17–34, 36–38, 41–43, 45–58].

## Uptake of CHT

The pooled proportion for the uptake of CHT among Chinese PDWH was 65% (95% CI: 57%–73%). Significant heterogeneity was observed between individual studies included in the analysis ($I^2$ = 99.6%, $P < 0.001$) (Fig 2).

## HIV prevalence among PDWH couples

Among Chinese PDWH couples, the pooled HIV prevalence was 28% (95%CI: 24%–32%) (Fig 3). Significant heterogeneity, observed between individual studies, was included in the analysis ($I^2$ = 99.1%, $P < 0.001$).

## Subgroup analysis

The subgroup analyses are shown in Table 2. The pooled uptake rate of CHT among pregnant women (76%, 95%CI: 66%–86%) was higher than that of MSM (49%, 95%CI: 30%–68%). Of the eight studies among MSM participants, three reported that their sexual partners were their spouses (legally married women). The uptake rate of CHT among PDWH couples (49%, 95% CI: 27%–70%) was similar to that among spouses (50%, 95%CI: 17%–83%). The CHT uptake rate in Yunnan province (69%, 95%CI: 59%–80%) was slightly higher than that in Guangxi province (65%, 95%CI: 49%–81%) (Table 2).

Subgroup analysis of HIV seroprevalence among CHT showed that the pooled prevalence among couples with HIV-infected pregnant women (53%, 95%CI: 27%–78%) was higher than that among couples of MSM (14%,95%CI: 10%–17%). Of the studies among MSM participants, two studies reported that their sexual partners were specifically their spouses. If these two studies were excluded, the pooled HIV prevalence was 13% (95% CI: 2%–24%) with moderate heterogeneity ($I^2$ = 40.7%, $P = 0.106$) (Table 2). The pooled HIV prevalence in Guangxi province (35%, 95%CI: 32%–39%) was slightly higher than that in Yunnan province (33%, 95%CI: 25%–42%), with moderate heterogeneity ($I^2$ = 49.2%, $P = 0.116$).

**Table 1. Study characteristics and outcomes of Chinese couples HIV testing (CHT).**

| No. | Publication | Province | Study design | Study period | Sample size[*] | Population | Type of union | Outcome measurement | CHT uptake (%)[**] | HIV prevalence among CHT (%)[***] | Quality score |
|---|---|---|---|---|---|---|---|---|---|---|---|
| 1 | Zheng [17], 2019 | Yunnan | CS | Jan 2012-Jun 2016 | 5086 | Pregnant women | Couples | Self-report | 81.3 | 32.7 | 8 |
| 2 | Yu [18], 2017 | Yunnan | CC | Jul 2012-Sep 2015 | 223 | Unspecified | Spouses | Self-report | 25.1 | NA | 8 |
| 3 | Lan [19], 2017 | Guangxi | CS | Until Nov 2016 | 405 | MSM | Spouses | Observation | 48.1 | NA | 8 |
| 4 | Zhao [20], 2017 | Jiangsu | CS | Until Dec 2015 | 158 | Unspecified | Couples | Observation | 74.7 | 18.6 | 8 |
| 5 | Wang X [21], 2015 | Sichuan, Yunnan, Xinjiang | CS | Jan 2012-Dec 2014 | 2007 | Pregnant women | Spouses | Self-report | 69.7 | 63.6 | 8 |
| 6 | Bai [22], 2016 | Yunnan | CS | Jan 2014-Dec 2015 | 263 | Unspecified | Spouses | Observation | NA | 30.4 | 8 |
| 7 | Li Q [23], 2016 | Yunnan | QE | May 2014-Dec 2015 | 105 | MSM | Couples | Observation | NA | 15.2 | 8 |
| 8 | Li [24], 2019 | Liaoning | RCT | Aug 2017-Jan 2019 | 94 | MSM | Couples | Observation | 17.0 | 26.3 | 8 |
| 9 | Chen [25], 2019 | Zhejiang | CS | Sep 2015-Sep 2016 | 321 | MSM | Couples | Observation | 41.1 | 13.8 | 8 |
| 10 | Xu [26], 2013 | Hebei | CS | Jan 1989-Dec 2011 | 232 | Unspecified | Couples or spouses | Observation | NA | 20.7 | 8 |
| 11 | Xu [27], 2014 | Yunnan | CS | Jan 1995-Dec 2013 | 2762 | Unspecified | Couples or spouses | Observation | 88.7 | 49.0 | 8 |
| 12 | Wang [28], 2018 | Jiangsu | CS | Jan 2010-Dec 2016 | 199 | MSM | Couples | Observation | 80.0 | 10.5 | 8 |
| 13 | Lian [29], 2019 | Fujian | CS | Jan 2015-Dec 2018 | 2937 | Unspecified | Spouses | Observation | 89.9 | 20.5 | 7 |
| 14 | Da [30], 2019 | Hubei | CS | Jan 2013-Dec 2017 | 2772 | MSM | Spouses | Observation | 28.9 | 18.7 | 8 |
| 15 | Li Y [31], 2016 | Yunnan | QE | May 2014-Dec 2015 | 118 | MSM | Couples | Observation | 60.2 | 13.1 | 8 |
| 16 | Liu [32], 2018 | Shanxi | CS | Until Nov 2015 | 246 | Unspecified | Couples or spouses | Observation | NA | 24.0 | 8 |
| 17 | Wang M [33], 2015 | Guangdong | CS | Jan 2010-Dec 2012 | 213 | Unspecified | Spouses | Observation | 82.2 | 41.1 | 8 |
| 18 | Hu [34], 2014 | Guangxi | CS | Aug 2012-Dec 2013 | 425 | Unspecified | Couples or spouses | Self-report | 70.4 | 40.5 | 8 |
| 19 | Zhu [35], 2010 | Yunnan | CS | Jan 1990-Sep 2009 | 196 | Unspecified | Spouses | Observation | NA | 52.0 | 6 |
| 20 | Zhong [36], 2016 | Guangxi | CS | Jan 2015-Dec 2015 | 45 | Unspecified | Couples or spouses | Observation | NA | 26.7 | 7 |
| 21 | Chen [37], 2018 | Anhui | CS | Jan 2000-Aug 2016 | 231 | Unspecified | Spouses | Observation | NA | 20.3 | 8 |
| 22 | Duan [38], 2004 | Yunnan | CS | March 2003 | 84 | Unspecified | Spouses | Observation | NA | 19.0 | 8 |
| 23 | Xi [39], 2009 | Yunnan | CS | Jan 1996-Dec 2008 | 88 | Unspecified | Couples or spouses | Observation | 83.0 | 31.5 | 6 |
| 24 | Xu [40], 2011 | Beijing | CS | NA | 451 | Unspecified | Spouses | Observation | 84.7 | 7.3 | 6 |

(*Continued*)

**Table 1.** (Continued)

| No. | Publication | Province | Study design | Study period | Sample size* | Population | Type of union | Outcome measurement | CHT uptake (%)** | HIV prevalence among CHT (%)*** | Quality score |
|---|---|---|---|---|---|---|---|---|---|---|---|
| | | | | | | | | | | | Study characteristics / Outcomes |
| 25 | Zhu [41], 2014 | Guangxi | CS | NA | 409 | Unspecified | Spouses | Observation | NA | 34.7 | 8 |
| 26 | Chen J [42], 2018 | Guangxi | CS | Jan 2006-Dec 2015 | 1658 | Unspecified | Spouses | Observation | NA | 34.1 | 8 |
| 27 | Chen [43], 2015 | Fujian | CS | Jan 2008-Dec 2013 | 872 | Unspecified | Couples or spouses | Observation | NA | 26.5 | 8 |
| 28 | Nong [44], 2019 | Guangxi | CS | Before Apr 2014 | 1307 | Unspecified | Couples or spouses | Observation | 76.3 | NA | 6 |
| 29 | Yang [45], 2018 | Jiangxi | CS | Jan 2017-Dec 2017 | 765 | Unspecified | Spouses | Observation | NA | 31.5 | 8 |
| 30 | Yang [46], 2019 | Jiangxi | RCT | Jan 2018-Dec 2017 | 206 | Unspecified | Couples or spouses | Observation | 50.0 | NA | 8 |
| 31 | Wang [47], 2008 | Shandong | CS | Jan 2003-Jun 2007 | 62 | Unspecified | Spouses | Observation | 66.1 | 39.0 | 8 |
| 32 | Zhang [48], 2015 | Shanghai | CS | Jul 1998-Jul 2014 | 307 | Unspecified | Couples or spouses | Self-report | 73.9 | 32.5 | 8 |
| 33 | Zeng [49], 2010 | Sichuan | CS | Jan 2008-March 2008 | 226 | Unspecified | Spouses | Observation | NA | 25.7 | 8 |
| 34 | Zhang [50], 2013 | Xinjiang | CS | Aug 2010-Feb 2011 | 383 | Unspecified | Couples or spouses | Observation | NA | 39.4 | 8 |
| 35 | Li J [51], 2016 | Yunnan, Henan, Sichuan, Guangxi, Xinjiang | CS | Jan 2011-Dec 2014 | 48931 | Unspecified | Spouses | Observation | NA | 24.6 | 8 |
| 36 | Lian [52], 2018 | Beijing, Jiangsu, Shanxi, Chongqing, Zhejiang, Hubei | CS | Apr 2014-Dec 2015 | 829 | MSM | Couples | Observation | NA | 11.0 | 8 |
| 37 | Chen S [53], 2018 | Hebei | CS | Jan 1995-Dec 2015 | 285 | Blood transfusion recipients | Spouses | Observation | NA | 20.8 | 7 |
| 38 | Lin [54], 2010 | Zhejiang | CS | May 2008-Mar 2010 | 129 | Unspecified | Couples or spouses | Observation | NA | 47.3 | 7 |
| 39 | Li J [55], 2017 | Unknown | CS | Jan 2014-Jun 2015 | 5081 | MSM | Spouses | Observation | 73.1 | 7.6 | 7 |
| 40 | Fu [56], 2016 | Zhejiang, Yunnan | QS | June 2014-May 2015 | 275 | MSM | Couples | Observation | NA | 10.5 | 8 |
| 41 | Mi [57], 2015 | Sichuan | QS | Dec 2008-Sep 2009 | 160 | MSM | Couples | Self-report | 45.6 | 25.6 | 8 |
| 42 | Qiu [58], 2009 | Yunnan | CS | Jul 2005-Jun 2006 | 28 | Pregnant women | Spouses | Observation | 78.6 | 63.6 | 7 |

NA, no data available; CS, cross-sectional; RCT, randomized controlled trial; QE, quasi-experimental study; CC, case-control study; PDWH, people diagnosed with HIV; Unspecified means no specific classification on the population of PDWH; MSM, men who have sex with men.

*Sample size was based on the number of PDWH.

** CHT uptake was calculated by the formula: (number of PDWH couples who had HIV testing) / (number of PDWH).

*** HIV prevalence among CHT (%) was calculated by the formula: (number of infected couples)/(number of couples who had HIV testing). Couples were defined as people in an ongoing sexual relationship. Spouses were defined as people who have a legal marital relationship.

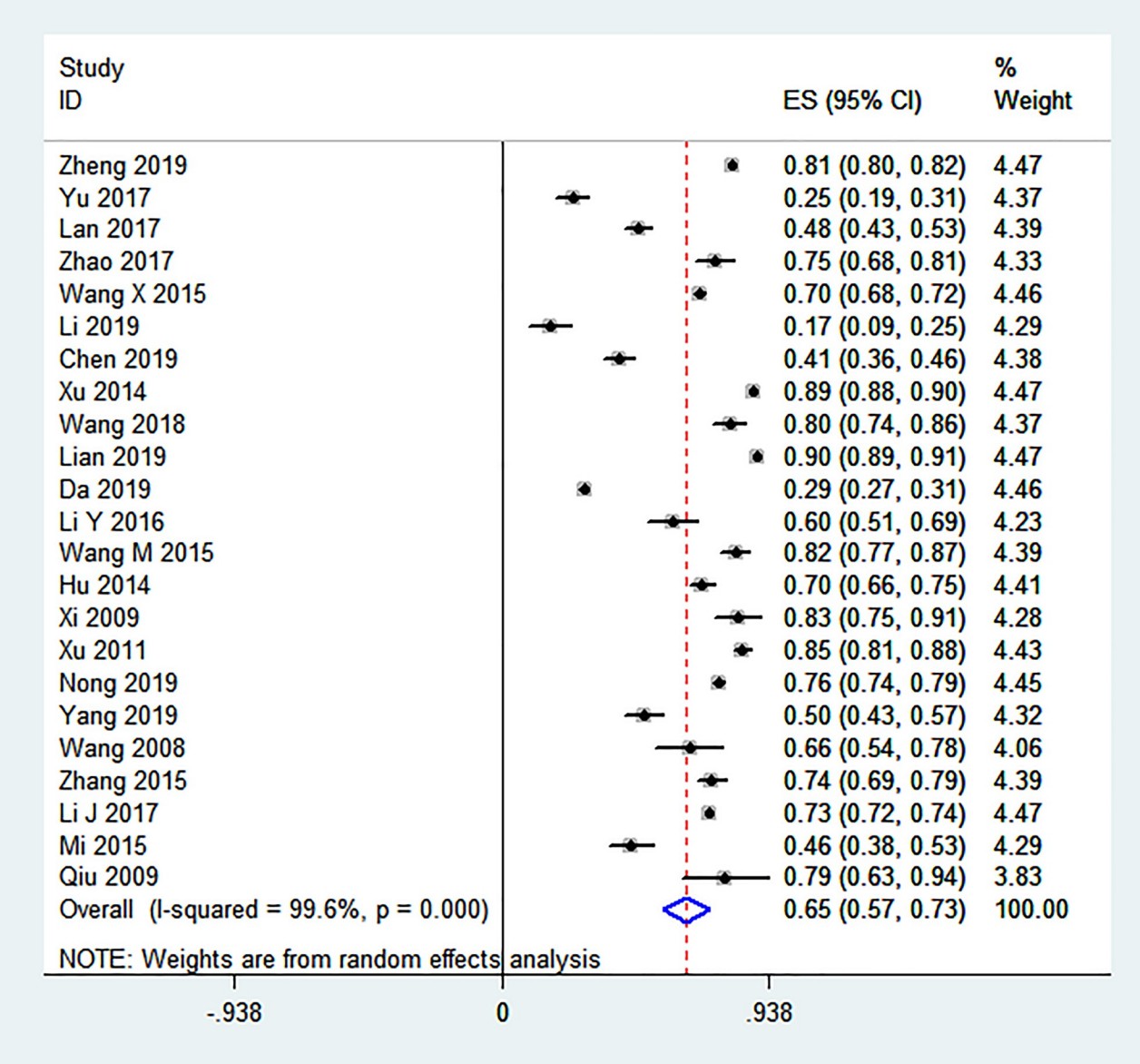

**Fig 2. Forest plot of CHT uptake among Chinese PDWH.**

## Publication bias

For the meta-analyses of CHT uptake rate and HIV prevalence among partners of PDWH, both Egger's ($t = -1.74$, $P = 0.097$; $t = 0.91$, $P = 0.369$) tests found no statistically significant difference, which indicates that there was no publication bias. However, results from the funnel plot of CHT uptake rate among PDWH couples showed that there might be missing studies at the bottom right of the graph (Fig 4), while results from the funnel plot of HIV prevalence among PDWH couples indicated that there might be missing studies from the bottom left of the graph (Fig 5).

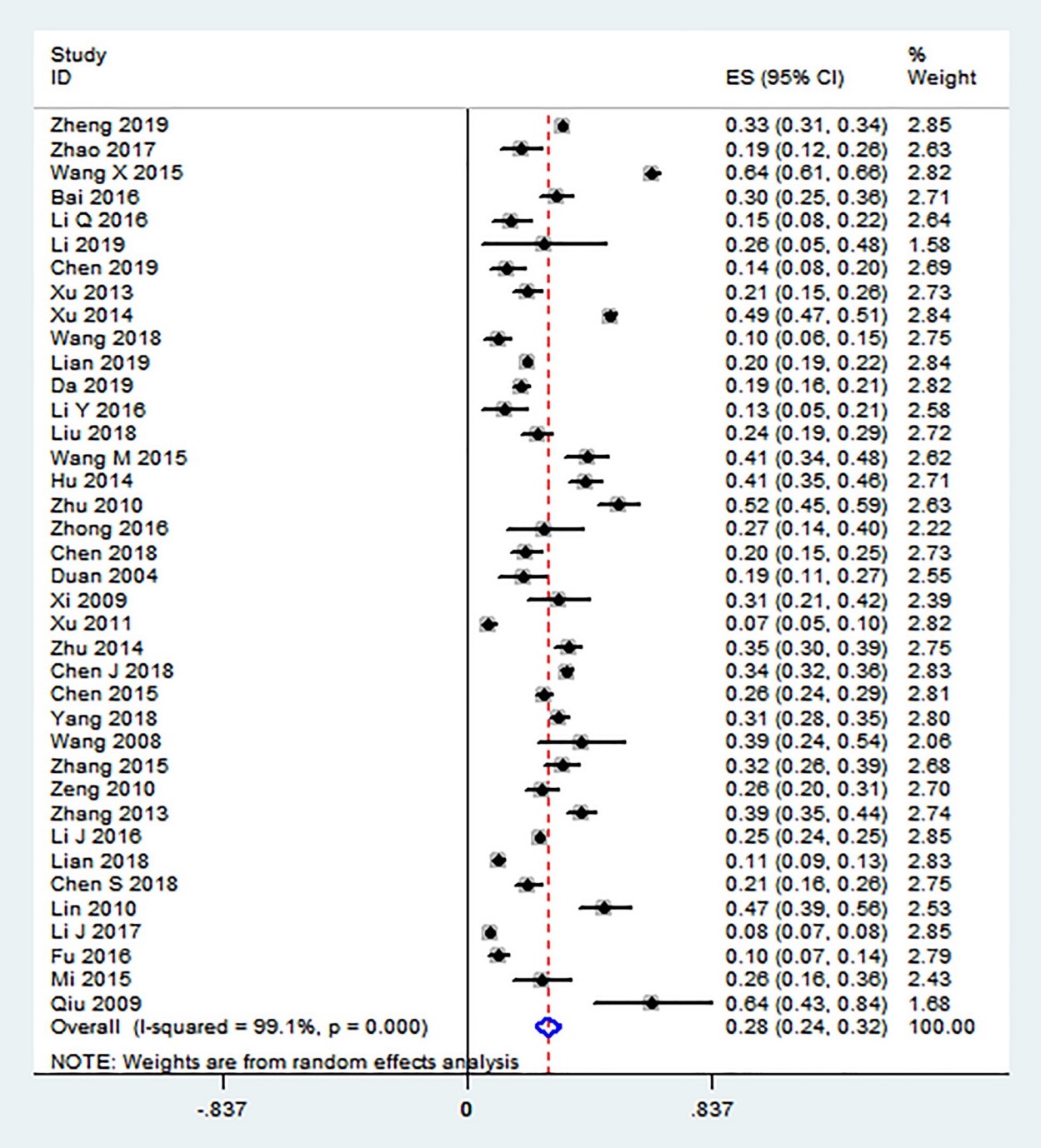

**Fig 3. Forest plot of HIV prevalence among couples of Chinese PDWH.**

## Sensitivity analysis

Sensitivity analyses indicated that none of the included studies significantly changed the pooled estimates for either study outcomes (Figs 6 and 7).

**Table 2. Subgroup analyses of uptake rate of CHT and HIV seroprevalence among PDWH couples.**

| Characteristic | Study, No. | ES (95% CI) | $I^2$, % | P value for heterogeneity |
|---|---|---|---|---|
| Uptake rate of CHT by study population | | | | |
| Pregnant women | 3 | 0.76 (0.66–0.86) | 98.0 | <0.001 |
| MSM | 8 | 0.49 (0.30–0.68) | 99.6 | <0.001 |
| Uptake rate of CHT by type of partners | | | | |
| Couples | 5 | 0.50 (0.17–0.83) | 98.0 | <0.001 |
| Spouses | 3 | 0.49 (0.27–0.70) | 99.9 | <0.001 |
| Uptake rate of CHT by province* | | | | |
| Yunnan | 6 | 0.69 (0.59–0.80) | 99.0 | <0.001 |
| Guangxi | 3 | 0.65 (0.49–0.81) | 98.1 | <0.001 |
| HIV prevalence in CHT by study population | | | | |
| Pregnant women | 3 | 0.53 (0.27–0.78) | 99.5 | <0.001 |
| MSM | 10 | 0.14 (0.10–0.17) | 89.2 | <0.001 |
| HIV prevalence in CHT by type union among MSM study participants | | | | |
| Couples | 8 | 0.13 (0.10–0.15) | 40.9 | 0.106 |
| Spouses | 2 | 0.13 (0.02–0.24) | 98.3 | <0.001 |
| HIV prevalence in CHT by province** | | | | |
| Yunnan | 9 | 0.33 (0.25–0.42) | 97.3 | <0.001 |
| Guangxi | 4 | 0.35 (0.32–0.39) | 49.2 | 0.116 |

CHT, couples' HIV testing; PDWH, people diagnosed with HIV; MSM, men who have sex with men.

* One study was excluded for analysis [21], which only reported the total uptake rate of CHT from several provinces.

** Three studies were excluded for analysis [21, 35, 40], which only reported total HIV prevalence in CHT from several provinces.

## Discussion

This meta-analysis provided pooled estimates of the uptake rate of CHT and HIV prevalence among Chinese PDWH couples, and presented the data according to study population, type of couple, and province. The CHT uptake rate was 65% among PDWH in China and 49% among couples of HIV-infected MSM. The results suggested that there were gaps in HIV testing among discordant sexual partners. The meta-analysis showed a pooled HIV prevalence of 28% among the PDWH couples in China. Our results highlighted the long-way PDWH couples have to go in order to achieve the Chinese government's goal to reduce HIV transmission rates between discordant spouses to below 1% by 2030 [11].

In the WHO guidelines for the "Partner Notification Policy," partner notification relied on the PDWH themselves to notify their sexual partners and receive CHT services [59]. While, taking HIV voluntary counseling and testing is voluntary for PDWH couples across most of China [60], partner notification in the four provinces of Yunnan, Henan, Zhejiang, and Gansu is mandatory among serodiscordant spouses [60]. The WHO guidelines also recommended the promotion of CHT by advocating HIV self-testing (HIVST) among high-risk populations and PDWH couples [59]. Few studies have shown the effectiveness of improved CHT uptake through the distribution of HIVST kits to sexual partners by antenatal and postpartum women [61]. The Chinese government also encouraged the implementation of HIVST and integrated it into routine HIV testing services [11]. However, there is no evidence based on rigorously designed studies that have explored the effects of HIVST on CHT uptake among PDWH in China.

CHT uptake rate is reasonably high among pregnant women study participants (76%), where the purpose may have been the prevention of mother-to-child transmission [62]. The

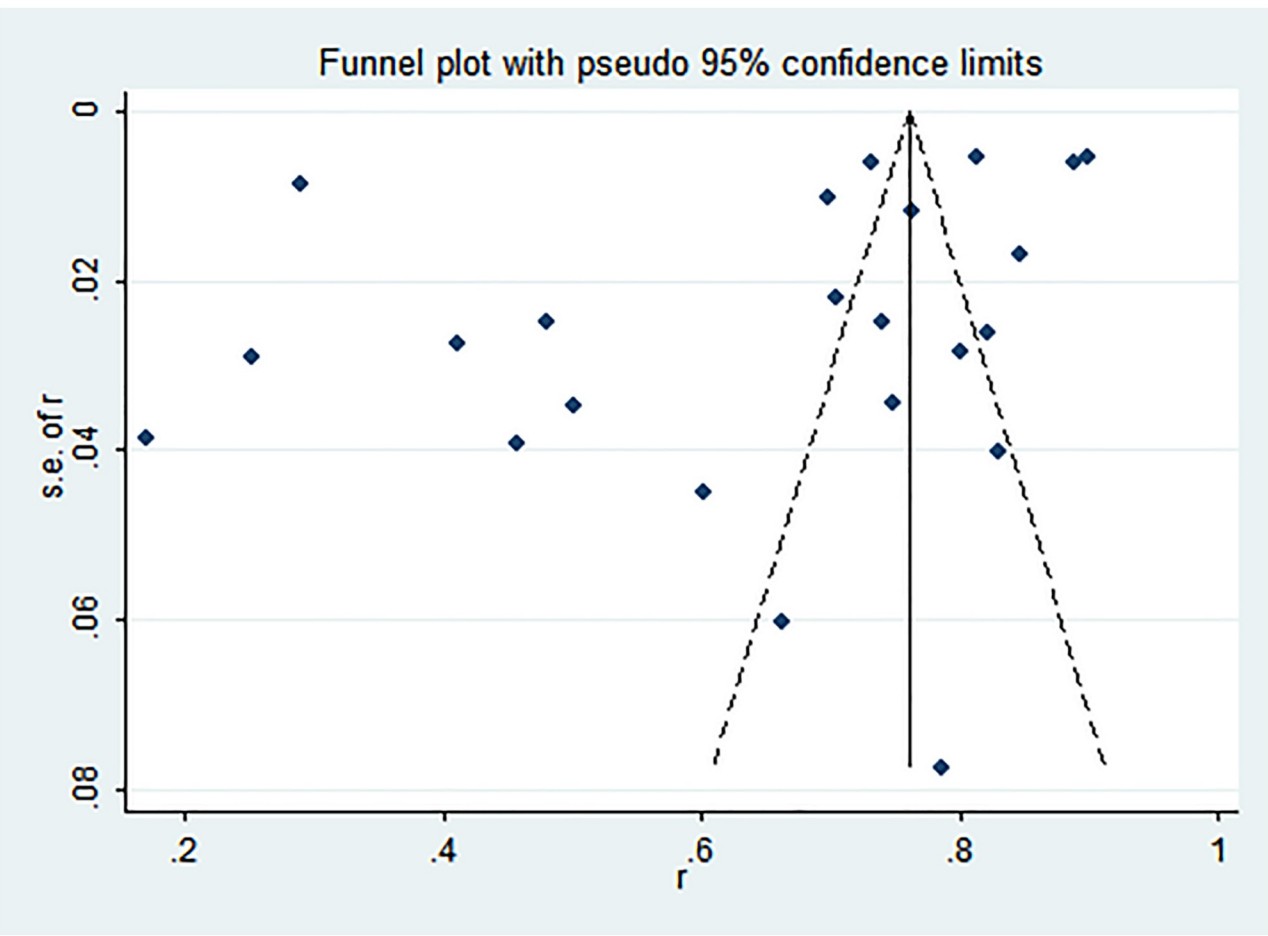

**Fig 4. Funnel plot of CHT uptake rate among PDWH couples.**

uptake rate was only 49% among HIV-infected MSM study participants. HIV-infected Chinese MSM may have a low rate of disclosure to their sexual partners because of the high levels of stigma and discrimination [59]. In addition, married MSM may have been concerned about the negative consequences of disclosing their sexual orientation to their female spouses [19]. These may have also led to a low CHT uptake. The CHT uptake in Guangxi province (65%) was similar to the pooled CHT uptake rate, but slightly lower than that in Yunnan province (69%). This may be due to the mandatory CHT policy among serodiscordant spouses in Yunnan province [60].

The pooled HIV prevalence in CHT was 28%, which is much higher than that among key populations in China, including MSM, injecting drug users, and sex workers [63, 64]. It is suggested that promoting CHT could be an efficient strategy to identify new infections. HIV prevalence among couples of infected pregnant women (53%) was 3.79 times higher than that among couples of infected MSM (14%). Pregnant women are not typically regarded as a high-risk population and consistent condom use is low with their partners. A recent study showed that 69% of pregnant women and their couples reported inconsistent condom use [65]. HIV prevalence rates among couples and spouses of MSM were similar. The implication may be that the wives of MSM could also be a high-risk population for HIV infection, since up to 70%

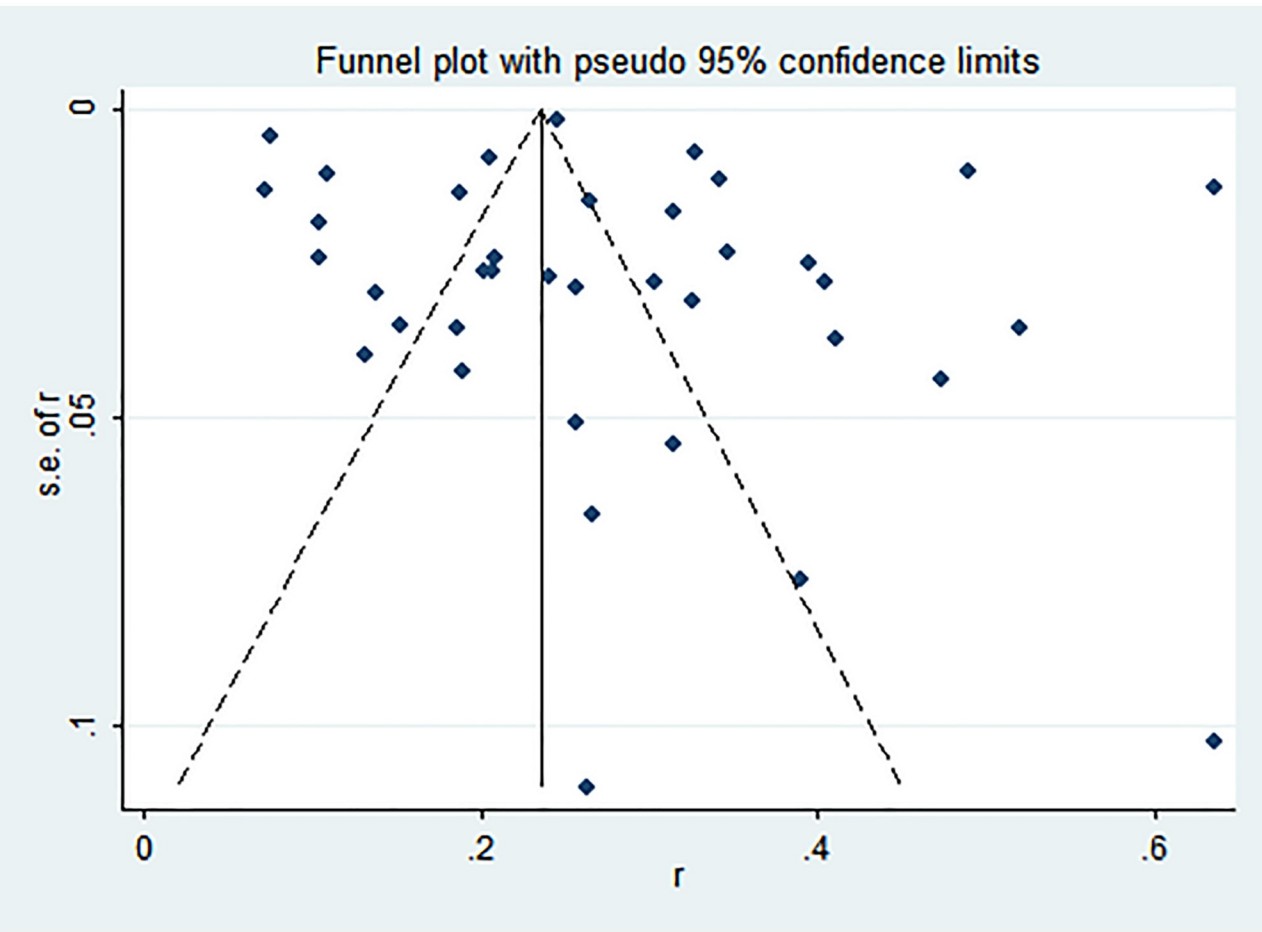

**Fig 5. Funnel plot of HIV seroprevalence among PDWH couples.**

of Chinese MSM would get married with women under the "filial piety" culture belief [66]. HIV prevalence rates among CHT in Yunnan (33%) and Guangzi (35%) provinces were similar, but much higher than the pooled HIV prevalence (28%). The main reason may be that both provinces have the highest HIV prevalence in China [67].

## Limitations

This meta-analysis had several limitations. First, selection bias could not be ruled out because the languages of the included studies were limited to English and Chinese. Second, information bias was likely to have existed because 14.3% of outcomes were evaluated by the participants' self-reporting. Third, the heterogeneity across the included studies was high, which may account for publication bias and limit generalizability of the findings. The main reasons might be that samples were recruited from 20 provinces with diverse HIV prevalence and partner notification policies. In addition, some studies had small sample sizes, which might also contribute to the heterogeneity.

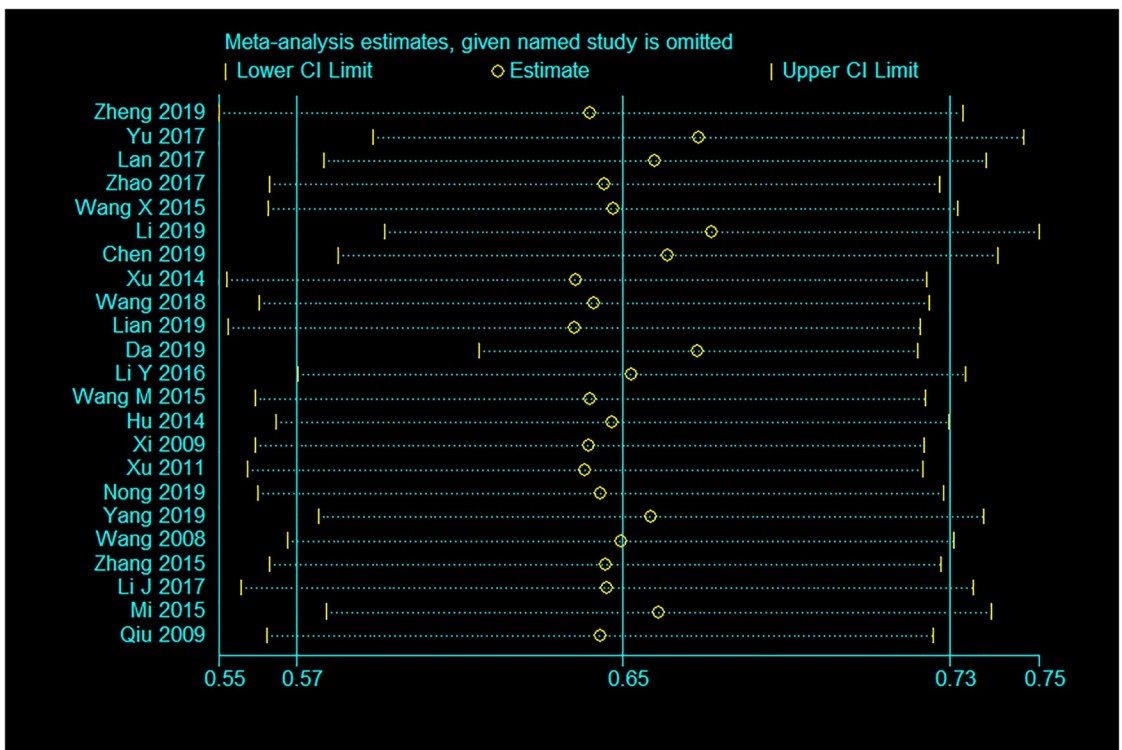

**Fig 6. Sensitivity analysis for pooled uptake rate of CHT.**

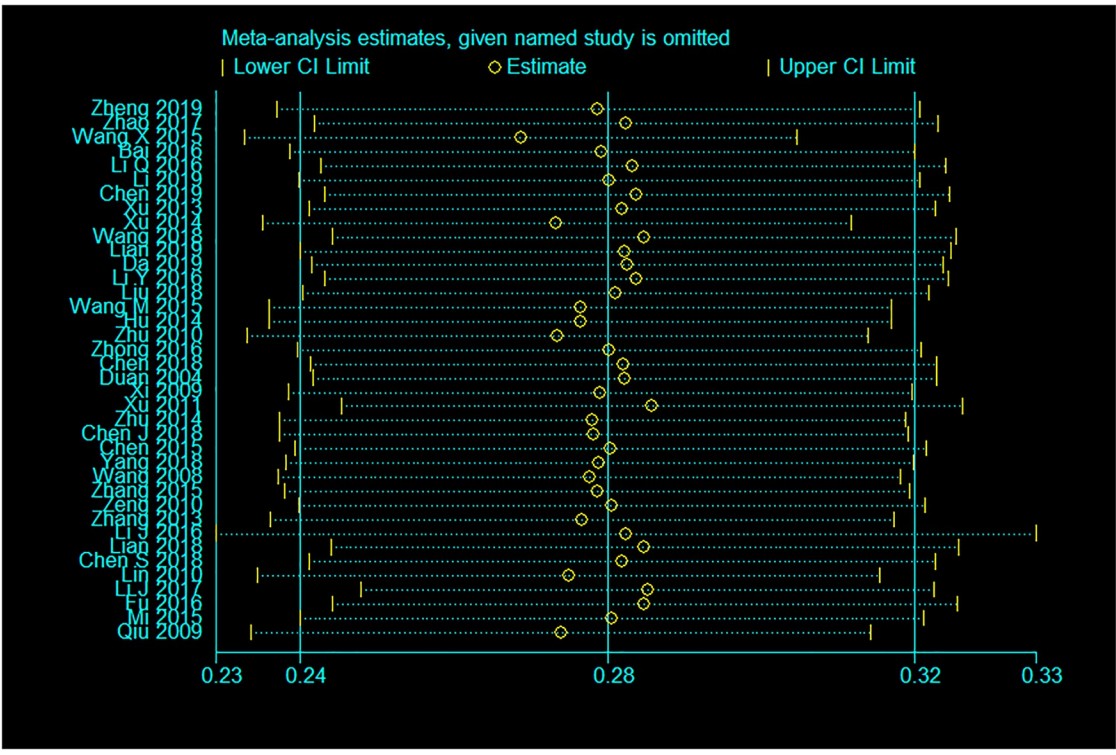

**Fig 7. Sensitivity analysis for HIV seroprevalence among PDWH couples.**

## Conclusions

Two-thirds of Chinese couples living with HIV have had an HIV test, of which 28% were positive. Couples of MSM had a lower HIV testing rate, which indicates that more effective strategies need to be carried out to improve couples' HIV testing among the Chinese MSM population.

## Supporting information

**S1 Table. PRISMA 2009 checklist.**
(DOC)

**S2 Table. Assessment of methodological quality of cross-sectional studies.**
(DOC)

## Author Contributions

**Conceptualization:** Ci Zhang, Honghong Wang, Xianhong Li.

**Data curation:** Ci Zhang.

**Formal analysis:** Ci Zhang.

**Funding acquisition:** Xianhong Li.

**Methodology:** Ci Zhang, Han-Zhu Qian, Xi Chen, Yan Shen, Honghong Wang, Xianhong Li.

**Project administration:** Xianhong Li.

**Software:** Ci Zhang.

**Supervision:** Honghong Wang, Xianhong Li.

**Validation:** Ci Zhang, Han-Zhu Qian, Xi Chen, Yan Shen, Honghong Wang, Xianhong Li.

**Visualization:** Ci Zhang, Han-Zhu Qian, Xi Chen, Scottie Bussell, Yan Shen, Honghong Wang, Xianhong Li.

**Writing – original draft:** Ci Zhang.

**Writing – review & editing:** Ci Zhang, Han-Zhu Qian, Xi Chen, Scottie Bussell, Yan Shen, Honghong Wang, Xianhong Li.

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
