## [Decision Letter · Decision Letter 0]

12 Oct 2020

PONE-D-20-11075

HIV testing and seroprevalence among couples of people living with HIV in China: a meta-analysis

PLOS ONE

Dear Dr. Xianhong Li,

Thank you for submitting your manuscript to PLOS ONE. After careful consideration, we feel that it has merit but does not fully meet PLOS ONE’s publication criteria as it currently stands. Therefore, we invite you to submit a revised version of the manuscript that addresses the points raised during the review process.

We look forward to receiving your revised manuscript.

Kind regards,

Qigui Yu, M.D./Ph.D

Academic Editor

PLOS ONE

Reviewers' comments:

Reviewer's Responses to Questions

**Comments to the Author**

1. Is the manuscript technically sound, and do the data support the conclusions?

Reviewer #1: Yes

Reviewer #2: Yes

2. Has the statistical analysis been performed appropriately and rigorously? 

Reviewer #1: Yes

Reviewer #2: I Don't Know

3. Have the authors made all data underlying the findings in their manuscript fully available?

Reviewer #1: Yes

Reviewer #2: Yes

4. Is the manuscript presented in an intelligible fashion and written in standard English?

Reviewer #1: No

Reviewer #2: No

5. Review Comments to the Author

Reviewer #1: First Read:

The paper is well written and does a great job reporting all necessary components of a systematic review and meta-analysis. The research questions and purpose are clear and straightforward. The search strategy and review methodology were clearly described and appropriate, however the manuscript needs to be reviewed for grammar and syntax, preferably by a native English speaker to improve clarity of your prose. Below are some of my concerns and questions.

General comments and questions:

To examine CHT and HIV prevalence among Chinese living with HIV - isn't this 100% prevalence? To be accurate should this be HIV prevalence among Chinese people. Or are you hoping to estimate prevalence of HIV among Chinese people who do not know their status? Or maybe discordant couples?

From further reading, I am assuming Chinese PLWH = Serodiscordant Chinese couples -> This needs to be better clarified and I have pointed it out in a few places, but this needs to be used consistently throughout the paper

Abstract:

It would be helpful to know how many studies attributed to each of the analysis with the estimates, it would clarify why MSM is stated to have 10 studies, however table 2 shows 8 studies for Uptake rate and 10 for HIV prevalence.

Line 21-22: "…among couples of Chinese people living with HIV…" - it is unclear whether or not both have HIV (seroconcordant) or only one person has HIV (serodiscordant). Assuming it is the latter, this needs to be written to be explicitly and clear throughout the paper.

Line 28-29: "We conducted three studies among…" - this is confusing, are there words missing? Should it be, "We conducted a subgroup analysis…"?

Line 36-37: "Almost two-thirds of Chinese couples living with HIV have had an HIV test, of which 28% were positive" - this implies that of the two-thirds that’s were tested, 28% were HIV positive. Were the 28% HIV-positive among the 65% who were tested or are these estimates separately calculated?

Introduction:

Line 45-46: the goals of 90-90-90 are: 90% of all people living with HIV will know their HIV status-90% of all people with diagnosed HIV infection will receive sustained antiretroviral therapy-90% of all people receiving antiretroviral therapy will have viral suppression. Testing coverage is not specified in the goals.

Line 53-54: "…encouraging CHT among PLWH could identify additional HIV infected individuals…" This is a very confusing statement, I think PLWH is meant to be "undiagnosed" or "serodiscordant"? Similar language is used throughout the introduction and needs to be clarified.

Methods:

Line 70: "…the proportion of CHT among Chinese PLWH…" for clarification, what is the denominator in this? Is it Chinese people living with HIV or Chinese couples or serodiscordant couples? This is somewhat described in the footnote in table 1, but this should be included in the text

Line 101: what were some of the variables extracted from the included studies? Was timeframe of testing abstracted, lifetime vs recent testing? This could have implications for interpreting results.

Line 118-119: were the couples assumed to be sexually monogamous couples or was this information collected?

Line108: Can you describe what some of these 8 items are assessing? Even in the supporting documents the scoring sheet does not describe the questions.

Line 121: What is the threshold for determining if a study does impact the pooled estimate in a meaningful way

Results:

The Uptake of CHT and Proportion of HIV among PLWH couples could be presented in tabular format, maybe integrated with table 2? The figures themselves can be difficult to interpret for readers who are not familiar with forest plots.

Line 136: Can an additional analysis of CHT and HIV prevalence be conducted among the some of the larger provinces that had a lot of data reported for them? This would be interesting to see.

Line 178: did any of the subgroup analyses uncover any sources of heterogeneity?

Line 200: The Eggers test may not have shown statistical evidence of publication bias, however I must disagree about the assessment of the funnel plots. Figure 4 seems to indicate there may be missing studies at the bottom right of the graph. Figure 5 seems to indicate some missing studies from the bottom left of the graph. However, this may not be helpful as these tests and bias assessments do not work well when heterogeneity is high. I think justifying lack of evidence for bias could be from your comprehensive search (of published and gray literature) would better serve the paper rather than the statistical testing.

Discussion:

The couples are now described as discordant sexual partners - This is the language that should be used throughout the paper rather than referring to them as Chinese PLWH couples.

Are there any efforts to increase serodiscordant couple HIV testing for the Chinese population? Are there any outside China that could be adapted and used?

Another potential limitation is the generalizability of the findings, meaning how well are these Chinese discordant couples included in the studies representative of all Chinese serodiscordant couples?

Tables and figures:

The figures need to be labeled and annotated in order for readers to be able to understand the them as a standalone result.

Reviewer #2: Please review and edit the article to be clearer. While the paper did meet expectations laid out in the abstract and the conclusions were well-thought out and appropriate given the data, there were several areas where more concise/clear language could have been used.

6. PLOS authors have the option to publish the peer review history of their article (what does this mean?). If published, this will include your full peer review and any attached files.

Reviewer #1: **Yes: **Jeffrey S Becasen

Reviewer #2: No

NOTE: While revising your submission, please upload your figure files to the Preflight Analysis and Conversion Engine (PACE) digital diagnostic tool, https://pacev2.apexcovantage.com/. PACE helps ensure that figures meet PLOS requirements. To use PACE, you must first register as a user. Registration is free. Then, login and navigate to the UPLOAD tab, where you will find detailed instructions on how to use the tool. If you encounter any issues or have any questions when using PACE, please email PLOS at figures@plos.org. Please note that Supporting Information files do not need this step.

---

## [Author Response · Author response to Decision Letter 0]

30 Nov 2020

The responses to reviewers and editors are attached in the file "Responses to reviewers".

---

## [Decision Letter · Decision Letter 1]

16 Feb 2021

HIV testing and seroprevalence among couples of people diagnosed with HIV in China: A meta-analysis

PONE-D-20-11075R1

Dear Dr. Li

We’re pleased to inform you that your manuscript has been judged scientifically suitable for publication and will be formally accepted for publication once it meets all outstanding technical requirements.

Kind regards,

Qigui Yu, M.D./Ph.D

Academic Editor

PLOS ONE

Additional Editor Comments (optional):

Reviewers' comments:

Reviewer's Responses to Questions

**Comments to the Author**

1. If the authors have adequately addressed your comments raised in a previous round of review and you feel that this manuscript is now acceptable for publication, you may indicate that here to bypass the “Comments to the Author” section, enter your conflict of interest statement in the “Confidential to Editor” section, and submit your "Accept" recommendation.

Reviewer #2: All comments have been addressed

Reviewer #3: All comments have been addressed

2. Is the manuscript technically sound, and do the data support the conclusions?

Reviewer #2: Yes

Reviewer #3: Yes

3. Has the statistical analysis been performed appropriately and rigorously? 

Reviewer #2: Yes

Reviewer #3: Yes

4. Have the authors made all data underlying the findings in their manuscript fully available?

Reviewer #2: Yes

Reviewer #3: Yes

5. Is the manuscript presented in an intelligible fashion and written in standard English?

Reviewer #2: Yes

Reviewer #3: Yes

6. Review Comments to the Author

Reviewer #2: (No Response)

Reviewer #3: (No Response)

7. PLOS authors have the option to publish the peer review history of their article (what does this mean?). If published, this will include your full peer review and any attached files.

Reviewer #2: No

Reviewer #3: No

---

## [Editor Report · Acceptance letter]

10 Mar 2021

PONE-D-20-11075R1 

HIV testing and seroprevalence among couples of people diagnosed with HIV in China: A meta-analysis 

Dear Dr. Li:

I'm pleased to inform you that your manuscript has been deemed suitable for publication in PLOS ONE. Congratulations! Your manuscript is now with our production department. 

Kind regards, 

on behalf of

Dr. Qigui Yu 

Academic Editor

PLOS ONE